# The heat wave knowledge, awareness, practice and behavior scale: Scale development, validation and reliability

**Ugurcan Sayili**[1,2]*, **Esin Siddikoglu**[1], **Betul Zehra Pirdal**[3], **Abdulkerim Uygur**[1], **Ferdane Seyma Toplu**[1], **Gunay Can**[1]

**1** Department of Public Health, Cerrahpasa Faculty of Medicine, Istanbul University-Cerrahpasa, Istanbul, Türkiye, **2** Department of Biostatistics and Medical Informatics, Institute of Graduate Studies in Health Sciences, Istanbul University, Istanbul, Türkiye, **3** Sultangazi District Health Directorate, Istanbul, Türkiye

* ugurcan.sayili@iuc.edu.tr

**Data Availability Statement:** All relevant data are within the paper and its Supporting information files.

## Abstract

Heat waves are extreme weather and climate events that threaten public health by increasing morbidity and mortality. To reduce the health effects of heat waves, it is necessary to increase the knowledge level of the public, conduct awareness and protection activities and monitor these activity outcomes. The present study aimed to develop and validate a Turkish language scale of heat wave knowledge, awareness, practice and behavior for Turkish nationality. After item generation and creating dimensions, content validity analysis was performed. To evaluate the validity and reliability of the knowledge construct, the difficulty index, discriminant index and Kuder Richardson 20 (KR20) were used. The validity and reliability of the awareness, practice and behavior constructs were evaluated with exploratory and confirmatory factor analyses, and Cronbach's alpha was used. The 15 items had a good difficulty, discrimination index and KR20 in the knowledge construct. The 14 items were yielded in EFA; 13 items were retained in CFA, and Cronbach's alpha values of 0.878, 0.768, 0.855, and 0.858 were obtained for total items, practice, awareness and behavior, respectively. Eventually, a Turkish language scale was developed that is reliable and valid for assessing heat wave knowledge, awareness, practice and behavior.

## Introduction

Climate change has been described as the greatest global threat of the 21st century endangering people's lives and well-being [1]. The Intergovernmental Panel on Climate Change (IPCC), in the 6th assessment report published in 2021, stated that the average temperature increase will vary between 1.4 and 4.4°C until 2100 according to different emission scenarios to the best estimates [2]. The Copernicus Climate Change Service released new data showing that the last seven years globally were the seven warmest on record by a clear margin [3]. Climate change has been expected to increase the intensity, frequency, duration and severity of extreme weather events such as heat waves [4]. Heat waves are extreme climatic events in which the weather is extremely hot for a prolonged time period [5]. Although it is generally understood

**Funding:** The author(s) received no specific funding for this work.

**Competing interests:** The authors have declared that no competing interests exist.

to be a 'prolonged period of excessive heat', there is no common definition of a heat wave or temperature thresholds [6].

Heat waves are considered a public health problem, as they are associated with increased hospital emergency admission and heat-related morbidity and mortality [7, 8]. The health effects of heat waves range from sunburn and heat stress to kidney failure, heart attack and even death [6].

Over the past 30 years, studies have reported excess deaths due to heat waves. In Chicago in 1995, more than 700 deaths were reported due to the heat wave [9]. The devastating effects of the heat wave were greatest in 2003, and it is estimated that there were over 70,000 more than expected deaths across Europe, of which 15,000 were in France [10]. In 2010, Russia experienced a 44-day heat wave, also called a "mega heat wave", which is estimated to have caused approximately 11,000 excess deaths [11]. The World Health Organization (WHO) states that between 1998 and 2017, over 166,000 people lost their lives due to heat waves [12]. In 2021, it was reported that there were 434 heat dome deaths compared to 80 community death expectations (a 440% increase) in Vancouver, Canada [13]. A study from China found that 5% excess deaths caused by heat waves between 2006–2011 [14].

The Mediterranean region, including Türkiye, is considered one of the most sensitive regions to climate change, especially because of the increase in temperature, precipitation changes and seasonal temperature variability. Diffenbaugh et al. reported that extreme temperatures are 2 to 5 times more common in the Mediterranean region [15].

There are limited studies evaluating the health effects of heat waves in Türkiye. It is known that hospital emergency service admissions and total deaths increase during heat waves [8]. Between 2013 and 2017, three heat waves in Istanbul caused 419 excess deaths [16]; similarly, in another study, it was reported that 4,281 excess deaths occurred due to heat waves between 2004 and 2017 in Istanbul [17].

According to the health belief model, which evolved gradually in response to very practical public health concerns, an individual's behavior and practices are related to understanding the severity of the risk and perceiving the benefit of taking action to reduce the threat [18]. The adverse health effects of heat waves can be predicted and prevented by certain public health protection measures, such as to increase the knowledge and awareness of the general population. Individuals reflect this information on their behaviors and practices during heat waves [5].

The public's knowledge, awareness, behavior and practices toward risks are important factors for reducing the health impact of heat waves [19, 20]. Many conditions, such as age, gender, educational status, socioeconomic status, and chronic disease, can affect the public's knowledge and practices [19, 21].

In addition, knowledge of heatwave protection and individual preparedness for heatwaves can be enhanced through specific government activities. Various countries have National Heat Health Action Plans (NHHAPs) aimed at improving knowledge and awareness of heat waves among the general population and particularly among vulnerable groups [21]. Some steps are being taken in this regard in Türkiye as well. Although the National Program for Reducing the Adverse Effects of Climate Change on Health was developed in Türkiye in 2015, it is not comprehensive regarding heat waves [22]. In addition, the name of the Ministry of Environment and Urbanization was changed to the Ministry of Environment, Urbanization and Climate Change in 2021. National heat-health plans should be established, the level of public knowledge, awareness, practice and behavior should be determined, and protection and prevention strategies should be acted upon rapidly.

To our knowledge, this study is the first developed scale for heat wave knowledge, awareness, practice and behavior in the Turkish language, and the second scale developed worldwide;

only one scale in Malaysia was developed, which is based on knowledge, risk perception, attitude and practice questionnaires on heat waves [23].

The present study aimed to develop and validate a Turkish language scale for heat wave knowledge, awareness, practice and behavior for Turkish nationality.

# Method

## Study design and population

This methodological study was conducted between February 2022 and June 2022. The survey data were collected via the LimeSurvey platform. Participants were reached via WhatsApp and Facebook.

The study was conducted in accordance with the Declaration of Helsinki. The eligibility criteria were age ≥18 years, Turkish citizenship, literacy, and acceptance to participate in the study.

For the sample size, in the guidelines, the general suggestion is a ratio of approximately 5 to 10 subjects per item up to approximately 300 subjects. Because of the knowledge structure included 20 items, and the awareness, behavior and practice structures included 15 items, the sample size was determined as 300 participants. Data collection ended with 308 participants. (Fig 1).

## Questionnaire development

**Item generation and content validity assessment.**   A literature review was carried out for item generation and creating constructs for scale by the authors. In this stage, with the literature review, 89 items were created for the 3 constructs of the pool scale (knowledge, awareness, behavior).

The items for the knowledge construct have 3 options (if it was a positive statement, 1: True, 0: False, 0: Not sure; if it was a negative, 1: False, 0: True, 0: Not sure). The scale for awareness and behavior was set on a 5-point Likert scale (if it was a positive statement, 1: strongly disagree 2: disagree, 3: unsure, 4: agree, 5: strongly agree; if it was a negative statement, 1: strongly agree 2: agree, 3: unsure, 4: disagree, 5: strongly disagree).

After the items were determined, the qualitative content validity was made with an expert panel method, and the pool scale was created; also, there was no suggestion to remove it from the item pool.

The invited three experts in the panel method were senior professors (environmental health and public health) or consultants (climate change) with academic titles in their research field. After the qualitative content validity with the panel method, the quantitative content validity evaluation phase has begun.

Quantitative content validity was analyzed with the content validity ratio (CVR) and content validity index (CVI). E-mail or telephone messages were sent to 30 experts for quantitative content validity. A total of 46.7% (n = 14) of the 30 experts whose opinions were consulted responded within 1 week. A reminder e-mail was sent to those who did not respond. In the second week, 6 (37.5%) experts made a return. A total of 20 expert feedback was received. These experts were specialists in medicine (public health, environmental health, emergency services, cardiologist, and family medicine) or environmental sciences (climatologist, environmental engineers). To determine the CVR, the experts were asked their opinions using a three-point scale (appropriate, appropriate but not necessary, unnecessary) for each item. However, 4 out of 20 experts chose it to be appropriate for similar questions (negatively and positively) but also for almost all (87–89 items). These four expert opinions were excluded. As a result, content validity analyses were conducted with the opinions of 16 experts.

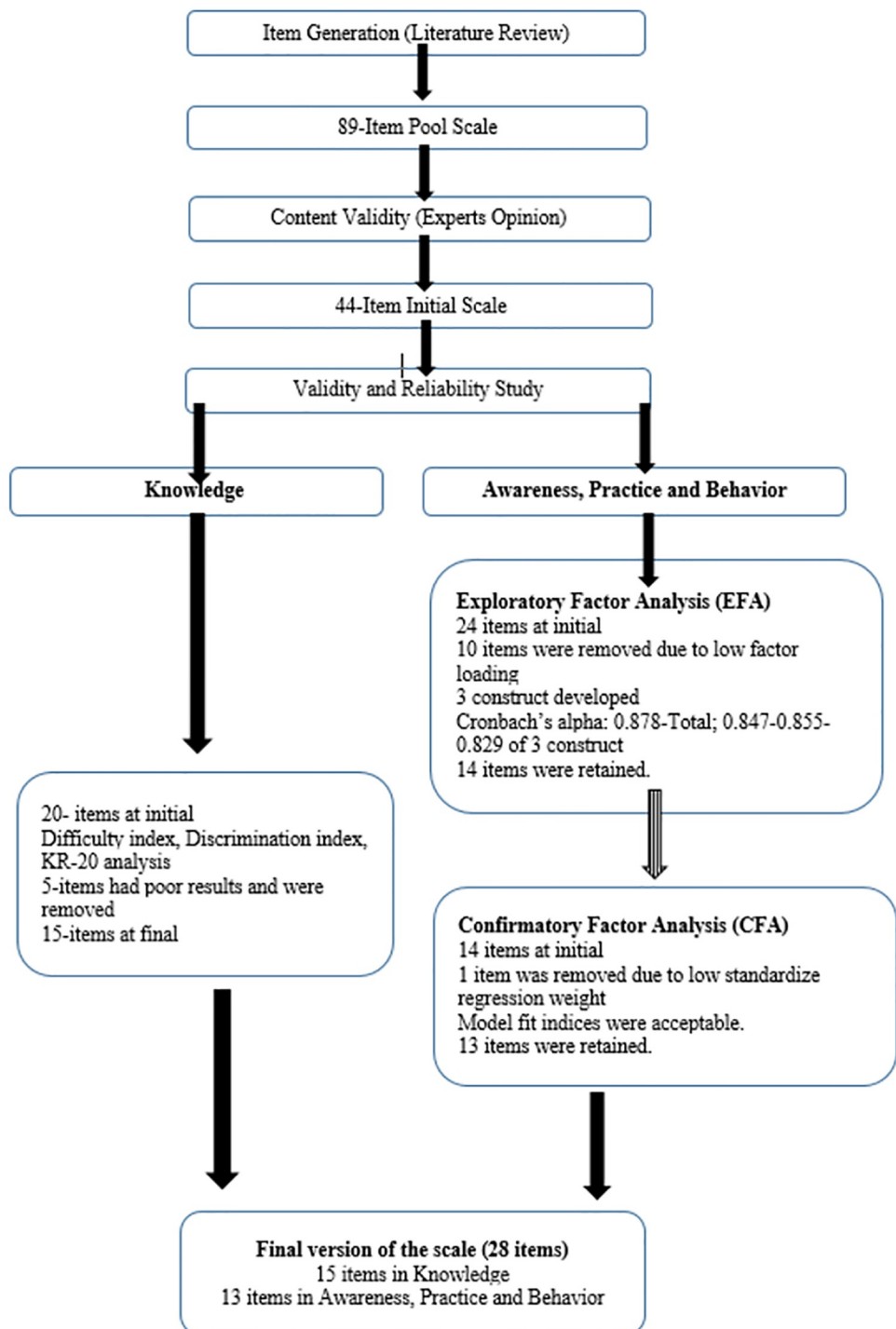

**Fig 1. Flowchart of heat wave knowledge, awareness, practice and behavior scale development, validity and reliability.**

The following formula was applied to determine the CVR.

$$\text{CVR} = \frac{n_e - \frac{N}{2}}{\frac{N}{2}}$$

In the formula, N is the total number of experts, and $n_e$ is the number of experts who selected the essential item.

The Lawshe technique was used to evaluate CVR. Based on the Lawshe technique, a minimum CVR of 0.500 and above was considered appropriate for an evaluation with 16 experts (p<0.05) [24].

The content validity index (CVI) was evaluated after CVR. For the CVI, a four-point scale was applied to the participants for each item, considering its simplicity, intelligibility and relevance to the subject. The CVI was calculated using the formula below:

$$\text{CVI} = \frac{\textit{The number of experts giving a rating of '3' or '4'}}{\textit{Total number of experts}}$$

The criterion for content validity of items was set. Items were considered adequate if there was >79% agreement, questionable if there was 70–79% agreement and unacceptable if there was <69% agreement [25].

After the content validity analyses for the pool scale, the initial scale to be applied to the participants was determined.

**Construct validity.** For the knowledge construct, the difficulty index ($p$) and discrimination index ($DI$) were calculated. The total scores of all participants were calculated, and then the upper and lower 27% points of respondents were identified.

The $p$ and $DI$ were calculated as follows:

$$p = [(H + L)/N] \times 100,$$

$$DI = 2 \times [(H - L)/N]$$

where "N" is the total number of participants in both the high and low groups. "H" and "L" are the number of correct responses in the high and low groups, respectively.

There are studies in the literature suggesting values of 30–70 or 20–90 for $p$ [26, 27]. Although we initially thought of accepting the range of 30–70, there were items in the 70–80 score range for the risk factors that the researchers considered important. Therefore, we accepted that the 30–80 point range is appropriate. Items with $p$ between 30–80 and $DI > 0.24$ were considered 'ideal' [26].

Exploratory factor analysis (EFA) and confirmatory factor analysis (CFA) were applied to demonstrate construct validity for the awareness, practice and behavior constructs. It was targeted to determine items and constructs with using EFA. After EFA, it was tested the confirmation of constructs and items. For EFA and CFA, the study sample was divided into two groups by simple randomization via SPSS. In EFA, the principal component method was used for factor extraction, and it is accepted that eigenvalues are greater than 1. Direct oblimin rotation was used in EFA. Items with factor loading<0.4 or factor loading difference <0.1 with loading to two factors or if only one/two items loading one factor were removed; others were retained. A Kaiser–Meyer–Olkin (KMO) value >0.6 and Bartlett sphericity test at p < 0.01 indicated that the data met the criteria for factor analysis [28].

The chi-square/df (CMIN/DF), comparative fit index (CFI), goodness of fit (GFI), incremental fit index (IFI), root mean square error of approximation (RMSEA) and standardized root mean square residual (SRMR) were used in CFA to compute the scale model fitness.

**Reliability.** For the knowledge construct, the Kuder-Richardson Formula 20 (KR-20) was used because it was coded as 0–1. For awareness, practice and behavior constructs, Cronbach's alpha was used. A Cronbach's alpha or KR-20 > 0.6 is considered acceptable [29]. Cristobal et al. suggested that corrected item-total correlation values lower than 0.30 are not acceptable [30].

## Ethics

This study was approved by the Istanbul University-Cerrahpasa, Non-Invasive Clinical Research Ethics Committee (10.03.2022–334078; 2022/26). When the survey link was clicked, a page was displayed that introduced the study and included the informed consent form. Participants who clicked the "I agree to participate in the study" button were able to access the page containing the survey and scale.

## Statistical analysis

The Statistical Package for the Social Sciences version 21.0 for Windows (IBM Corp., Armonk, NY, USA), JASP 0.14.1.0 and Microsoft Office Excel were used for data evaluation and analysis. Categorical variables are presented as frequencies (n) and percentages (%), and numeric variables are presented as the mean ± standard deviation and median (interquartile range (IQR)) values. For two response items, the discrimination index and difficulty index were calculated to evaluate validity, and the Kuder-Richardson 20 value was calculated to evaluate reliability. For 5-point Likert items, exploratory and confirmatory factor analysis and Cronbach's alpha, Spearman's correlation analysis were used to evaluate validity and reliability.

# Results

## Demographic characteristics

Table 1 represents the participants' characteristics. A total of 308 people participated in the study, and most of the participants were female (60.7%). The mean participant's age was 38.5 ±15.2. Most of the participants were 18–35 years old (42.5%), were married (55.7%), had children (52.6%), and had a university degree (80.2%). Most of the participants said, "Income equals expense" (49.2%); 26.9% of the participants said "income less than expenses"; 23.9% of the participants said "income more than expenses". A total of 21.2% of the participants had a chronic disease. A total of 33.7% of the participants had air conditioning in their home. The most common cooling method used was opening windows or balconies (61.1%). The AFA and CFA study groups were similar in terms of basic characteristics (Table 1).

## Content validity

The pool scale consisted of 89 items. A total of 44 items had CVR≥0.50 and CVI>%79. As a result, in the content validity analyses, 44 items that met the CVR≥0.50 and CVI>79% criteria remained on the scale. Of the 44 items, 20 belonged to the knowledge construct, and 24 belonged to the awareness and behavior construct.

## Validity and reliability of knowledge construct

Of the 20 items in the knowledge construct, 5 items were removed due to inappropriate values in the difficulty index or discrimination index. (K3, K4, K14, K17, K19). The 20 items and 15 items in the knowledge construct had a good reliability index (pKR20: 0.629 and 0.629, respectively). As a result, the 15-item knowledge construct was evaluated as valid and reliable (Table 2).

**Table 1. Demographic characteristics of participants in the study (n = 308).**

| Characteristics | All group (n:308) | CFA sample (n:154) | EFA sample (n:154) | p value |
|---|---|---|---|---|
| **Gender** | | | | |
| Female | 187(60.7%) | 87(56.5%) | 100(64.9%) | 0.129‡ |
| Male | 121(39.8%) | 67(43.5%) | 54(35.1%) | |
| **Age** (Mean±std) | 38.5±15.2 | 38.8±14.8 | 38.2±15.7 | 0.741* |
| **BMI** (Mean±std) | 25.6±4.9 | 25.8±4.6 | 25.3±5.1 | 0.309* |
| **Age group** | | | | |
| 18–34.9 | 131(42.5%) | 64(41.6%) | 67(43.5%) | |
| 35–44.9 | 44(14.3%) | 21(13.6%) | 23(14.9%) | |
| 45–54.9 | 59(19.2%) | 32(20.8%) | 27(17.5%) | 0.917‡ |
| 55–64.9 | 34(11.0%) | 18(11.7%) | 16(10.4%) | |
| ≥65 | 16(5.2%) | 7(4.5%) | 9(5.8%) | |
| Missing | 24(7.8%) | 12 (7.8%) | 12 (7.8%) | |
| **Marital Status** | | | | |
| Married | 171(55.7%) | 94(61.0%) | 77(50.3%) | |
| Single | 128(41.7%) | 58(37.7%) | 70(45.8%) | 0.093† |
| Others | 8(2.6%) | 2(1.3%) | 6(3.9%) | |
| **Having child** | | | | |
| No | 145(47.4%) | 68(44.4%) | 77(50.3%) | 0.303‡ |
| Yes | 161(52.6%) | 85(55.6%) | 76(49.7%) | |
| **Education Status** | | | | |
| Primary School | 10(3.2%) | 6(3.8%) | 4(2.5%) | |
| High school | 51(16.6%) | 27(17.5%) | 24(15.6%) | 0.452† |
| University | 210(68.2%) | 99(64.3%) | 111(72.1%) | |
| Graduate | 37(12.0%) | 22(14.3%) | 15(9.7%) | |
| **Income Status** | | | | |
| Income less than expenses | 82(26.9%) | 38(24.7%) | 44(29.1%) | |
| Income equals expense | 150(49.2%) | 81(52.6%) | 69(45.7%) | 0.474‡ |
| Income more than expense | 73(23.9%) | 35(22.7%) | 38(25.2%) | |
| **Having chronic disease** | | | | |
| No | 242(78.8%) | 118(76.6%) | 124(81.0%) | 0.343‡ |
| Yes | 65(21.2%) | 36(23.4%) | 29(19.0%) | |
| **Having air conditioning** | | | | |
| No | 203(66.3%) | 94(61.4%) | 109(71.2%) | 0.070‡ |
| Yes | 103(33.7%) | 59(38.6%) | 44(28.8%) | |
| **The most common cooling method** | | | | |
| Air conditioning | 54(17.5%) | 27(17.5%) | 27(17.5%) | |
| Fan | 53(17.2%) | 26(16.9%) | 27(17.5%) | 0.992‡ |
| window/balcony | 188(61.1%) | 94(61.0%) | 94(61.0%) | |
| Others | 13(4.2%) | 7(4.5%) | 6(3.9%) | |
| **Individuals over 65 at home** | | | | |
| No | 250(81.7%) | 124(81.0%) | 126(82.4%) | 0.767‡ |
| Yes | 56(18.3%) | 29(19.0%) | 27(17.6%) | |

‡: Chi-square test;

†:Fisher's exact test;

*: Independent sample t test was applied.

**Table 2. Validity and reliability analysis for knowledge construct.**

| Item Number | Difficulty index | Discrimination index | Status |
|---|---|---|---|
| K1 | 77.71 | 0.25 | Appropriate |
| K2 | 61.45 | 0.55 | Appropriate |
| **K3** | **88.55** | **0.13** | **Removed** |
| **K4** | **37.95** | **0.23** | **Removed** |
| K5 | 40.96 | 0.34 | Appropriate |
| K6 | 68.07 | 0.49 | Appropriate |
| K7 | 66.87 | 0.35 | Appropriate |
| K8 | 75.90 | 0.29 | Appropriate |
| K9 | 57.23 | 0.42 | Appropriate |
| K10 | 67.47 | 0.51 | Appropriate |
| K11 | 77.71 | 0.25 | Appropriate |
| K12 | 78.31 | 0.29 | Appropriate |
| K13 | 78.92 | 0.30 | Appropriate |
| **K14** | **88.55** | **0.23** | **Removed** |
| K15 | 55.42 | 0.63 | Appropriate |
| K16 | 53.01 | 0.48 | Appropriate |
| **K17** | **84.34** | 0.29 | **Removed** |
| K18 | 73.49 | 0.43 | Appropriate |
| **K19** | **88.55** | **0.23** | **Removed** |
| K20 | 61.45 | 0.60 | Appropriate |

First trial for KR20: 0.629

After inappropriate items were removed, KR20: 0.629

## Validity and reliability of awareness, practice and behavior construct

**Exploratory factor analysis.** EFA was applied for the validity of awareness and behavior constructs. The KMO index was 0.881; Bartlett's test of sphericity was found to be significant; and the total variance explained was 62.08% with a 3-factor construct (Table 3).

Although it was designed as an awareness and behavior construct at the beginning of the study, a 3-factor structure was found in the EFA. The items designed as the behavior construct were divided into two different constructs. It was evaluated that some of them predicted behavior and some of them predicted practice; therefore, they were not excluded from the scale and kept in the scale. A total of 10 items were removed due to low factor loading (<0.4), loading two factors with factor loading difference <0.1 or only 1–2 item with loading one factor. The first factor represents "practice" for 41.7% of variance; the second factor "awareness" explained 12.2% of variance, and the third factor "behavior" explained 8.2% of variance (Table 4).

**Confirmatory Factor Analysis (CFA).** After exploratory factor analysis, confirmatory factor analysis was performed in a different sample. One item (B2) was removed for having a

**Table 3. Kaiser-Meyer-Olkin test, Bartlett's test and total variance explained results for exploratory factor analysis.**

| Kaiser-Meyer-Olkin Measure of Sampling Adequacy | | 0.881 |
|---|---|---|
| Bartlett's Test of Sphericity | p value | <0.001 |
| Total Variance Explained | 3 Component | %62.08 |

**Table 4. Results of exploratory factor analysis for awareness, practice and behavior construct (n:154).**

| Construct | Item | Factor Loading | | |
| --- | --- | --- | --- | --- |
| | | Factor 1 | Factor 2 | Factor 3 |
| **Practice** | **B4** | 0.74 | | |
| | **B5** | 0.72 | | |
| | **B6** | 0.74 | | |
| | **B7** | 0.69 | | |
| | **B9** | 0.55 | | |
| **Awareness** | **A1** | | 0.79 | |
| | **A2** | | 0.87 | |
| | **A3** | | 0.73 | |
| | **A7** | | 0.82 | |
| | **A10** | | 0.63 | |
| **Behavior** | **B10** | | | 0.87 |
| | **B11** | | | 0.90 |
| | **B12** | | | 0.65 |
| | **B2** | | | 0.45 |

low standardized regression weight (0.300). After the item was removed, the scale was evaluated as appropriate according to CFA. All items have >0.5 standardize regression weights. The model has good fit indices such as CMIN/DF, CFI, IFI, RMSEA, SRMR and GFI. The CMIN/DF was 2.081, the CFI was 0.924, the IFI value was 0.925, the RMSEA value was 0.084, the SRMR value was 0.066 and the GFI value was 0.886 in this study. Since the model fit values met the acceptable criteria, the model was considered appropriate. Table 5 and Fig 2 represent the results of confirmatory factor analysis results.

**Reliability analysis.** The subscales and scales have good reliability results. Reliability analysis indicated Cronbach's alpha values of 0.855, 0.768, 0.858 and 0.878 (awareness, practice, behavior and total items, respectively). All items had item-total correlation coefficients greater than 0.50 and corrected item-total correlation coefficients greater than 0.4 (Table 6). Table 7 represents the correlation of the constructs. The correlation coefficients between knowledge, awareness, practice and behavior were statistically significant.

**Table 5. Standardized regression weights on confirmatory factor analysis (n:154).**

| Component | Item | ß (Standardize estimate) | $R^2$ |
| --- | --- | --- | --- |
| **F1 (Practice)** | **B4** | 0.642 | 0.412 |
| | **B5** | 0.702 | 0.493 |
| | **B6** | 0.685 | 0.469 |
| | **B7** | 0.508 | 0.258 |
| | **B9** | 0.513 | 0.264 |
| **F2 (Awareness)** | **A1** | 0.785 | 0.616 |
| | **A2** | 0.853 | 0.728 |
| | **A3** | 0.537 | 0.288 |
| | **A7** | 0.742 | 0.550 |
| | **A10** | 0.733 | 0.538 |
| **F3 (Behavior)** | **B10** | 0.855 | 0.731 |
| | **B11** | 0.817 | 0.667 |
| | **B12** | 0.876 | 0.767 |

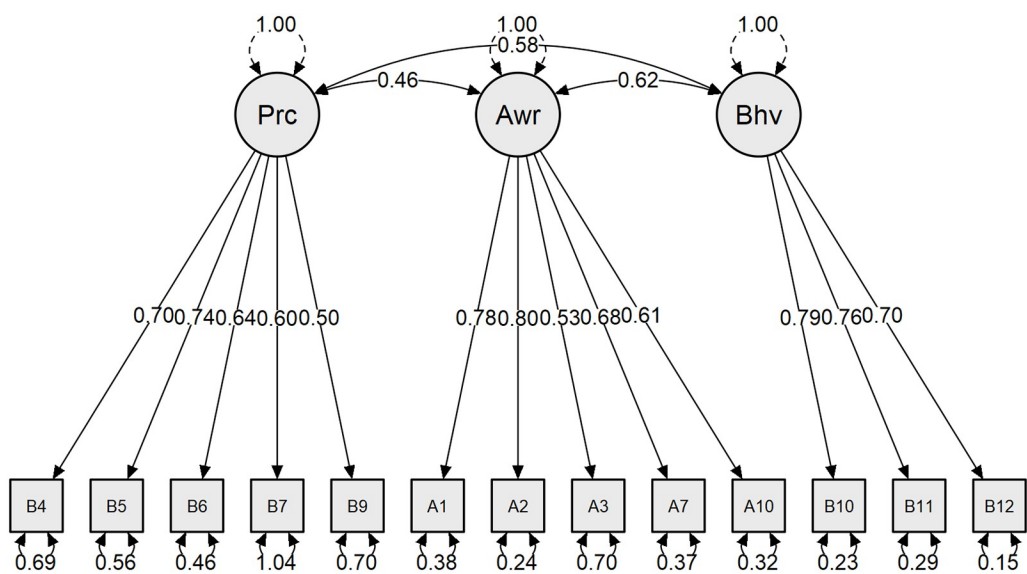

**Fig 2. CFA results of awareness, practice and behavior construct (Prc: Practice; Awr: Awareness; Bhv: Behavior).**

**Table 6. The reliability results for awareness, practice and behavior construct (n: 308).**

| Construct | Item | Reliability Analysis | | | |
|---|---|---|---|---|---|
| | | Item-Total Correlation | Corrected Item-Total Correlation | Cronbach's Alpha | |
| | | | | (Construct) | (Total) |
| Practice | B4 | 0.62 | 0.54 | 0.768 | 0.878 |
| | B5 | 0.62 | 0.56 | | |
| | B6 | 0.64 | 0.61 | | |
| | B7 | 0.52 | 0.43 | | |
| | B9 | 0.52 | 0.43 | | |
| Awareness | A1 | 0.55 | 0.57 | 0.855 | |
| | A2 | 0.54 | 0.62 | | |
| | A3 | 0.55 | 0.51 | | |
| | A7 | 0.50 | 0.59 | | |
| | A10 | 0.55 | 0.64 | | |
| Behavior | B10 | 0.71 | 0.65 | 0.858 | |
| | B11 | 0.61 | 0.54 | | |
| | B12 | 0.66 | 0.64 | | |

**Table 7. The correlations of the knowledge, awareness, practice and behavior constructs.**

| | | Awareness | Practice | Behavior |
|---|---|---|---|---|
| Knowledge | r | 0.41 | 0.24 | 0.25 |
| | p value | <0.001 | <0.001 | <0.001 |
| Awareness | r | | 0.32 | 0.39 |
| | p value | | <0.001 | <0.001 |
| Practice | r | | | 0.47 |
| | p value | | | <0.001 |

## Discussion

This study aimed to develop a knowledge, awareness, practice and behavior scale on heat waves that threaten public health and are increasingly common with global climate change.

This Turkish language scale on heat waves is the first valid and reliable scale in Türkiye and the second in the world. To our knowledge, only one in Malaysia was developed, which is on a knowledge, risk perception, attitude and practice questionnaire on heat waves. In this study, a valid and reliable scale was developed in the Turkish language for knowledge, awareness, practice and behavior about heat waves.

Many studies emphasize the need to increase knowledge for behavior and practice development [5, 19]. After the 2003 heat wave in France, it was found that the devastating health effects predicted in the 2006 heat wave decreased thanks to the increased awareness and policies implemented [31]. Kim et al. found that awareness was an important factor in health-protective behaviors to reduce the risk of heat waves [20]. Lefevre et al. stated that having heard recommendations on heat waves indicated having implemented more heat protection behaviors, and intending to implement more protection behaviors in future hot summers [32]. In our study, we found positively significant correlations between knowledge, awareness, practice and behavior. Many countries have developed national heat health plans and implemented heat wave warning systems. Increased knowledge and awareness are achieved with these implementations and programs. The effectiveness of these implementations and plans in reducing mortality is known [21, 33].

In general, it is recommended to begin with a pool of items that is three or four times as large as the final scale [34]. In this study, the 89 generated items were in the pool scale; the 44 items passed the content validity analysis, and the 28 items were retained in the final version of the scale.

The first construct of the scale was a knowledge construct that contains 20 items in the initial scale. There are studies in the literature suggesting values of 30–70 or 20–90 for the difficulty index ($p$) [26, 27]. In this study, items with $p$ between 30 and 80 were considered 'ideal'. Because the items with a $p$ of 70–80 were important to evaluate the knowledge of risky (vulnerable) groups such as children, elderly individuals and pregnant women. According to these criteria, 15 of 20 items were found to be appropriate. The KR-20 value was adequate for 15 item scales (KR-20:0.629). Similar to our results, Arsad et al. found that 16 of 20 items were appropriate in the heat wave knowledge construct [23].

In the EFA, the KMO value was 0.881, and Bartlett's test of sphericity was significant. This indicates that sampling is adequate and that the model is acceptable. Although the awareness and behavior construct was planned, a three-component construct was found. This 3 component explained 62.1% of the total variance. It appears that the behavior construct's items are split into two (behavior and practice). This splitting may be due to the characteristics of the Turkish language and the way of thinking in Turkish. Because "practice" and "behavior" have similar meanings in the Turkish language structure. In EFA, a total of 10 items were removed due to low factor loading (<0.4), loading two factors with factor loading difference <0.1 or one/two items with one factor loading. In addition, in EFA, one item (B2) showed a factor loading of 0.45, while the other 13 items were greater than 0.5. In the CFA, this item is removed because it has a low standardized regression weight (0.300). All other items have >0.5 standardize regression weights. The model has good fit indices such as CMIN/DF, CFI, IFI, RMSEA and GFI. The rule of thumb for (approximately) normed fit indices is that a value greater than 0.9, an RMSEA and SRMR less than 0.08 and CMIN/DF less than 3 indicates an acceptable fit [35]. Some studies suggested that RMSEA values in the range of 0.05 to 0.08 indicate fair fit and 0.08 to 0.10 indicate mediocre fit [36]. The CMIN/DF less than 3 is

acceptable, and it is 2.081 in the present study. For CFI and IFI values greater than 0.9 are acceptable. The CFI is 0.924 and the IFI value is 0.925 in this study, which is acceptable since it is greater than 0.9. The SRMR value was 0.066, which is acceptable. The RMSEA value was 0.084 and GFI value was 0.886 in this study, and they were close to recommended values. As a result, the model is good, with 3 components and 13 items.

Cronbach's alpha values of 0.855, 0.768, 0.858 and 0.878 (awareness, practice, behavior and 13 items, respectively) and all item-total correlation coefficients were greater than 0.50, indicating good reliability results in this study. In addition, the corrected item-total correlation coefficients were greater than 0.40, which is acceptable. The Cronbach alpha value, which is often used for internal consistency reliability, is between 0.60 and 0.70 are considered "acceptable in exploratory research," values between 0.70 and 0.90 range from "satisfactory to good" [37]. Arsad et al. showed a cronbach's alpha value varying from 0.78 to 0.84 and an item-total correlation coefficient varying from 0.30 to 0.60 in their scale on risk perception, attitude and practice for heat waves [23].

Lenzholner et al. reported that a general lack of awareness about urban climate adaptations. In addition, it is reported that media campaigns and good practice projects seem most effective in raising awareness [38]. Individual disaster preparedness requires appropriate knowledge and awareness how to behave and to practice for protect oneself [5]. In our study, it was found that knowledge, awareness, behavior and practice showed a statistically significant positive correlation with each other. Li et al. reported that a significant correlation between heat wave knowledge, awareness and practice [19]. Hajat et al provided a list for the evidence of the most commonly provided heat protection advice and advice on optimal clinical and public health practice that is expected to reduce heat-related health problems [39].

Eventually, this scale is a valid and reliable scale for evaluating knowledge, awareness, practice and behavior for heat waves. Although this scale was developed in Turkish, it actually sheds light on an important worldwide problem, and it can be applied in many different countries and cultures by making adaptation studies. As mentioned earlier, another scale developed similarly is in Malaysia [23]. Both two scales have knowledge and practice constructs, and awareness/Risk perception and attitude/behavior construct can be considered similar, as well as their question content.

A few limitations should be considered for this study. First, data were collected via an online survey, and the survey was distributed with WhatsApp and Facebook. Therefore, generalizability is limited. Although the results of this study confirm the psychometric properties of the scale, similar studies in Turkish and other communities are needed for generalizability. Second, all data were collected using self-reported measures. Third, a similar scale could not be applied, and convergent validity was not assessed. Because this was the first study in the Turkish language, and one of the limited studies in the world. This limitation can be actually a strength of the study. This study sheds light on future research.

## Conclusion

This study is the first developed scale for heat wave knowledge, awareness, practice and behavior in the Turkish language, and the second scale was developed worldwide (one in the Malay language). The heat wave knowledge, awareness, practice and behavior scale showed good validity and reliability results. The increasing frequency of heat waves is an important public health problem that increases mortality and morbidity. To protect against the negative effects of heat waves, it is necessary to develop different protection and prevention strategies. This newly developed scale can be applied in different cultures and countries by making adaptation researches and used by researchers to measure the level of knowledge, awareness, practice and

behavior regarding heat waves. In addition, information acquired with this scale can guide both researchers and policy makers in determining and monitoring the protection and prevention strategies to be developed.

## Supporting information

**S1 File. All items of pool scale.**
(DOCX)

**S2 File. The final version of valid and reliable scale.**
(DOCX)

**S3 File. Data.**
(SAV)

## Author Contributions

**Conceptualization:** Ugurcan Sayili.

**Data curation:** Esin Siddikoglu, Betul Zehra Pirdal, Abdulkerim Uygur, Ferdane Seyma Toplu.

**Formal analysis:** Ugurcan Sayili, Abdulkerim Uygur, Ferdane Seyma Toplu.

**Investigation:** Abdulkerim Uygur, Ferdane Seyma Toplu.

**Methodology:** Ugurcan Sayili, Esin Siddikoglu, Betul Zehra Pirdal, Gunay Can.

**Project administration:** Gunay Can.

**Supervision:** Gunay Can.

**Writing – original draft:** Ugurcan Sayili, Esin Siddikoglu, Betul Zehra Pirdal, Abdulkerim Uygur, Ferdane Seyma Toplu, Gunay Can.

**Writing – review & editing:** Ugurcan Sayili.

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
