## [Decision Letter · Decision Letter 0]

31 Oct 2022

PONE-D-22-26697The Heat Wave Knowledge, Awareness, Practice and Behavior Scale: Scale Development, Validation and ReliabilityPLOS ONE

Dear Dr. Sayili,

Thank you for submitting your manuscript to PLOS ONE. After careful consideration, we feel that it has merit but does not fully meet PLOS ONE’s publication criteria as it currently stands. Therefore, we invite you to submit a revised version of the manuscript that addresses the points raised during the review process.

We look forward to receiving your revised manuscript.

Kind regards,

Supat Chupradit, Ph.D., M.Ed., B.Sc.(OT), B.P.A., B.Ed., B.A.

Academic Editor

PLOS ONE

Journal Requirements:

Reviewers' comments:

Reviewer's Responses to Questions

**Comments to the Author**

1. Is the manuscript technically sound, and do the data support the conclusions?

Reviewer #1: Yes

Reviewer #2: Yes

Reviewer #3: Yes

Reviewer #4: Yes

Reviewer #5: Yes

2. Has the statistical analysis been performed appropriately and rigorously? 

Reviewer #1: No

Reviewer #2: Yes

Reviewer #3: Yes

Reviewer #4: Yes

Reviewer #5: Yes

3. Have the authors made all data underlying the findings in their manuscript fully available?

Reviewer #1: Yes

Reviewer #2: Yes

Reviewer #3: Yes

Reviewer #4: Yes

Reviewer #5: No

4. Is the manuscript presented in an intelligible fashion and written in standard English?

Reviewer #1: Yes

Reviewer #2: Yes

Reviewer #3: No

Reviewer #4: Yes

Reviewer #5: Yes

5. Review Comments to the Author

Reviewer #1: 1-According to using the CFA, the authors have decied to using the rules of thumb. Thus, the authors should shows the number of parameter estimate in this study because the rules of thumb calculate by 10 or 20 samples per 1 parameter.

2-This study found the theory framework is fit and related with empirical data. Therefore, the authors should describe the reasons why this model is fit and the measure of this study why there are suitable for using to study in the future study.

3-In the table 5, the authors should shows the raw beta and r-square statistic.

4-This study should concern in the indicators that illustrulate fit model. (might be concern with Hair et al., 2010 fit model statistic)

Reviewer #2: The article topic "The Heat Wave Knowledge, Awareness, Practice and Behavior Scale: Scale Development, Validation and Reliability" I think concise, interesting and good manuscript ready to publish. However, authors should recheck reference all update.

Reviewer #3: Manuscript title: The Heat Wave Knowledge, Awareness, Practice and Behavior Scale: Scale Development, Validation and Reliability. Overall interest, concise but I suggest some issue below.

- Please clarify why you use CFA for analysis. How about detail that important for you use?

- Check Table and Figure more clear.

- Update references that relate your literature and context.

Thank you

Reviewer #4: The author has provided a clear explanation of the study's findings, which are backed up with reliable source data and acceptable conclusions.

It is an important issue that the world recognizes and attaches importance to. Therefore, it would be helpful to develop an instrument to measure this issue.

Reviewer #5: The Abstract should be written as one paragraph and must not contain any sub-headings in this section.

What the authors should clearly show the readers is the details of the four measurement questions that the authors developed. Saying that it can be used only benefits the authors, not the readers. The academic contribution this paper should have is that it tells the readers how to measure four things related to heat waves.

In addition to the tools for Turkey, the authors should also make this paper useful to readers internationally, at least linking it to the pre-existing Malaysian scales.

However, the question the authors should also explain is why it is stated that this is a Turkish scale only, not a universal scale. What makes it unique and incompatible? Differences in social or cultural contexts?

6. PLOS authors have the option to publish the peer review history of their article (what does this mean?). If published, this will include your full peer review and any attached files.

Reviewer #1: No

Reviewer #2: No

Reviewer #3: No

Reviewer #4: No

Reviewer #5: **Yes: **Kittisak JERMSITTIPARSERT

---

## [Author Response · Author response to Decision Letter 0]

24 Nov 2022

Comments to the Author

Journal Requirements:

Response: The manuscript file is checked. It is corrected.

Response: This information was given in beginning of the methods section. However, in order to your comments, we relocated this paragraph into the ethics subsection.

“When the survey link was clicked, a page was displayed that introduced the study and included the informed consent form. Participants who clicked the "I agree to participate in the study" button were able to access the page containing the survey and scale.”

Response: This section is corrected. 

“All relevant data are within the paper and its Supporting Information files.”

Review Comments to the Author

Reviewer #1: 1-According to using the CFA, the authors have decied to using the rules of thumb. Thus, the authors should shows the number of parameter estimate in this study because the rules of thumb calculate by 10 or 20 samples per 1 parameter.

Response: We thank to reviewer’s valuable comment. We have detailed the sample size part in the text.

For the sample size, in the guidelines, the general suggestion is a ratio of approximately 5 to 10 subjects per item up to approximately 300 subjects. 

We have given this information in the method section.

Because of the knowledge structure included 20 items, and the awareness, behavior and practice structures included 15 items, the sample size was determined as 300 participants. 

2-This study found the theory framework is fit and related with empirical data. Therefore, the authors should describe the reasons why this model is fit and the measure of this study why there are suitable for using to study in the future study.

Response: We have made some correction in the introduction and discussion section.

Correction in the introduction

According to the health belief model, which evolved gradually in response to very practical public health concerns, an individual's behavior and practices are related to understanding the severity of the risk and perceiving the benefit of taking action to reduce the threat [17]. The adverse health effects of heat waves can be predicted and prevented by certain public health protection measures, such as to increase the knowledge and awareness of the general population. Individuals reflect this information on their behaviors and practices during heat waves. [5].

Correction in the discussion:

Individual disaster preparedness requires appropriate knowledge and awareness how to behave and to practice for protect oneself [5]. In our study, it was found that knowledge, awareness, behavior and practice showed a statistically significant positive correlation with each other.

3-In the table 5, the authors should shows the raw beta and r-square statistic.

Response: We thank to reviewer’s valuable comment. In the table 5, We have given the beta and r-square statistics.

4-This study should concern in the indicators that illustrulate fit model. (might be concern with Hair et al., 2010 fit model statistic)

Response: We have revised it.

The rule of thumb for (approximately) normed fit indices is that a value greater than 0.9, an RMSEA and SRMR less than 0.08 and CMIN/DF less than 3 indicates an acceptable fit [33]. Some studies suggested that RMSEA values in the range of 0.05 to 0.08 indicate fair fit and 0.08 to 0.10 indicate mediocre fit [34]. The CMIN/DF less than 3 is acceptable, and it is 2.081 in the present study. For CFI and IFI values greater than 0.9 are acceptable. The CFI is 0.924 and the IFI value is 0.925 in this study, which is acceptable since it is greater than 0.9. The SRMR value was 0.066, which is acceptable. The RMSEA value was 0.084 and GFI value was 0.886 in this study, and they were close to recommended values. 

The Cronbach alpha value, which is often used for internal consistency reliability, is between 0.60 and 0.70 are considered “acceptable in exploratory research,” values between 0.70 and 0.90 range from “satisfactory to good” [35].

Reviewer #2: The article topic "The Heat Wave Knowledge, Awareness, Practice and Behavior Scale: Scale Development, Validation and Reliability" I think concise, interesting and good manuscript ready to publish. However, authors should recheck reference all update.

Response: We thank to reviewer’s valuable comment. The literature was checked, and additions marked in the manuscript.

Reviewer #3: Manuscript title: The Heat Wave Knowledge, Awareness, Practice and Behavior Scale: Scale Development, Validation and Reliability. Overall interest, concise but I suggest some issue below.

Response: Thank you for your valuable comment. We hope that our article will take attention of the readers.

- Please clarify why you use CFA for analysis. How about detail that important for you use?

Response: Because while developing this scale, we first determined the structures ourselves and then the items suitable for these structures. When we started working, we had a structure in mind (which we expected). For this reason, we determined which of the items we determined with EFA explained the structure. Then, we checked the accuracy of the structure we determined in CFA.

We have added a sentence to methods section.

It was targeted to determine items and constructs with using EFA. After EFA, it was tested the confirmation of constructs and items.

- Check Table and Figure more clear.

Response: Table 4 can be confusing as it shows both exploratory factor analysis and reliability analysis results. Therefore, we divided Table4 into two and showed the reliability results in a different table. In addition, some minor word correction has been made in other tables. Also, we changed the AMOS output figure with JASP figure in CFA. If the reviewers have any concerns about other tables or figures and say specifically, we are ready to correct them.

- Update references that relate your literature and context.

Response: The literature was checked, and additions marked in the manuscript.

Reviewer #4: The author has provided a clear explanation of the study's findings, which are backed up with reliable source data and acceptable conclusions.

It is an important issue that the world recognizes and attaches importance to. Therefore, it would be helpful to develop an instrument to measure this issue.

Response: Thank you for your valuable comment. We hope that our article will take attention of the readers.

Reviewer #5: The Abstract should be written as one paragraph and must not contain any sub-headings in this section.

Response: We have corrected the abstract.

What the authors should clearly show the readers is the details of the four measurement questions that the authors developed. Saying that it can be used only benefits the authors, not the readers. The academic contribution this paper should have is that it tells the readers how to measure four things related to heat waves.

Response: We thank to reviewer for pointing out this. We have made some correction in the introduction and discussion section.

According to the health belief model, which evolved gradually in response to very practical public health concerns, an individual's behavior and practices are related to understanding the severity of the risk and perceiving the benefit of taking action to reduce the threat [17]. The adverse health effects of heat waves can be predicted and prevented by certain public health protection measures, such as to increase the knowledge and awareness of the general population. Individuals reflect this information on their behaviors and practices during heat waves. [5].

Individual disaster preparedness requires appropriate knowledge and awareness how to behave and to practice for protect oneself [5]. In our study, it was found that knowledge, awareness, behavior and practice showed a statistically significant positive correlation with each other. Li et al. reported that a significant correlation between heat wave knowledge, awareness and practice [18]. Hajat et al provided a list for the evidence of the most commonly provided heat protection advice and advice on optimal clinical and public health practice that is expected to reduce heat-related health problems [36].

In addition to the tools for Turkey, the authors should also make this paper useful to readers internationally, at least linking it to the pre-existing Malaysian scales.

However, the question the authors should also explain is why it is stated that this is a Turkish scale only, not a universal scale. What makes it unique and incompatible? Differences in social or cultural contexts?

Response: We thank to reviewer for pointing out this.

Since the scale was developed and implemented in Turkish, we thought it would be more accurate to specify it as a Turkish scale. Of course, the scale we have developed can be used by making adaptation studies in other countries, in this way it is actually universal. 

However, by addressing this comment of the referee, we provided an explanation for it in the last paragparh of discussion, and conclusions.

Although this scale was developed in Turkish, it actually sheds light on an important worldwide problem, and it can be applied in many different countries and cultures by making adaptation studies. As mentioned earlier, another scale developed similarly is in Malaysia [21]. Both two scales have knowledge and practice constructs, and awareness/Risk perception and attitude/behavior construct can be considered similar, as well as their question content.

---

## [Decision Letter · Decision Letter 1]

4 Dec 2022

The Heat Wave Knowledge, Awareness, Practice and Behavior Scale: Scale Development, Validation and Reliability

PONE-D-22-26697R1

Dear Dr. Sayili,

We’re pleased to inform you that your manuscript has been judged scientifically suitable for publication and will be formally accepted for publication once it meets all outstanding technical requirements.

Kind regards,

Supat Chupradit, Ph.D., M.Ed., B.Sc.(OT), B.P.A., B.Ed., B.A.

Academic Editor

PLOS ONE

Additional Editor Comments (optional):

Reviewers' comments:

Reviewer's Responses to Questions

**Comments to the Author**

1. If the authors have adequately addressed your comments raised in a previous round of review and you feel that this manuscript is now acceptable for publication, you may indicate that here to bypass the “Comments to the Author” section, enter your conflict of interest statement in the “Confidential to Editor” section, and submit your "Accept" recommendation.

Reviewer #2: All comments have been addressed

Reviewer #3: All comments have been addressed

Reviewer #5: All comments have been addressed

2. Is the manuscript technically sound, and do the data support the conclusions?

Reviewer #2: Yes

Reviewer #3: Yes

Reviewer #5: Yes

3. Has the statistical analysis been performed appropriately and rigorously? 

Reviewer #2: Yes

Reviewer #3: Yes

Reviewer #5: Yes

4. Have the authors made all data underlying the findings in their manuscript fully available?

Reviewer #2: Yes

Reviewer #3: Yes

Reviewer #5: Yes

5. Is the manuscript presented in an intelligible fashion and written in standard English?

Reviewer #2: Yes

Reviewer #3: Yes

Reviewer #5: Yes

6. Review Comments to the Author

Reviewer #2: manuscript title: The Heat Wave Knowledge, Awareness, Practice and Behavior Scale: Scale Development, Validation and Reliability. Its improve article base on referees comments. accept this manuscript revise version.

Reviewer #3: The manuscript title: The Heat Wave Knowledge, Awareness, Practice and Behavior Scale: Scale Development, Validation and Reliability. Overall, this paper has clear research issues, appropriate study methods, and presents interesting results.

Reviewer #5: This paper has been fairly completely updated based on the reviewer's suggestions. According to my opinion, the authors may improve it a bit as follows.

1) The abstract must be written in one paragraph only.

2) Move the sample's demographic data into the Research methods, not the Research results.

7. PLOS authors have the option to publish the peer review history of their article (what does this mean?). If published, this will include your full peer review and any attached files.

Reviewer #2: No

Reviewer #3: No

Reviewer #5: **Yes: **Kittisak JERMSITTIPARSERT

---

## [Editor Report · Acceptance letter]

13 Dec 2022

PONE-D-22-26697R1 

The heat wave knowledge, awareness, practice and behavior scale: Scale development, validation and reliability 

Dear Dr. Sayili:

I'm pleased to inform you that your manuscript has been deemed suitable for publication in PLOS ONE. Congratulations! Your manuscript is now with our production department. 

Kind regards, 

on behalf of

Assistant Professor Supat Chupradit 

Academic Editor

PLOS ONE